# Modeling Dual-Drive Gantry Stages with Heavy-Load and Optimal Synchronous Controls with Force-Feed-Forward Decoupling

**DOI:** 10.3390/e24081153

**Published:** 2022-08-19

**Authors:** Hanjun Xie, Qinruo Wang

**Affiliations:** Automation College, Guangdong University of Technology, Guangzhou 510000, China

**Keywords:** dual-drive gantry stage, synchronization control, GSLQR optimal control, convex optimization, force-FF decoupling, virtual centroid, heavy-load

## Abstract

The application of precision dual-drive gantry stages in intelligent manufacturing is increasing. However, the loads of dual drive motors can be severely inconsistent due to the movement of heavy loads on the horizontal crossbeam, resulting in synchronization errors in the same direction movement of dual-drive motors. This phenomenon affects the machining accuracy of the gantry stage and is an critical problem that should be immediately solved. A novel optimal synchronization control algorithm based on model decoupling is proposed to solve the problem. First, an accurate physical model is established to obtain the essential characteristics of the heavy-load dual-drive gantry stage in which the rigid-flexible coupling dynamic is considered. It includes the crossbeam’s linear motion and rotational motion of the non-constant moment of inertia. The established model is verified by using the actual system. By defining the virtual centroid of the crossbeam, the cross-coupling force between dual-drive motors is quantified. Then, the virtual-centroid-based Gantry Synchronization Linear Quadratic Regulator (GSLQR) optimal control and force-Feed-Forward (FF) decoupling control algorithm is proposed. The result of the comparative experiment shows the effectiveness and superiority of the proposed algorithm.

## 1. Introduction

Among all configurations of large span, long-stroke Cartesian robot systems, the industrial dual-drive gantry, also known as a dual-drive H-gantry(DHG), has attracted increasing attention from industry and academia [1,2]. The application demands for precision dual-drive gantry stages is increasing in many fields, such as laser cutting/welding applications, lithography, placement machines, solar panels, LCD panels, CNC machining centers, precision metrology, and CT scanning. In the DHG structure, two permanent magnetic linear motors (PMLM) are mounted on two parallel guide rails to push the gantry crossbeam in the same direction. Heavy-loads (such as the laser cutting head) are mounted on the crossbeam and driven by a single PMLM. The advantage of this structure is that the stage could obtain higher rigidity and power density.

Various factors will cause the dynamic characteristics of the dual-drive motors to be inconsistent, such as heavy-load motions on the crossbeam, varying damping characteristics of the dual-drive motors, and time-varying thermal-mechanical properties. It will lead to the synchronization error of the linear movement of dual-drive motors and the small-angle rotation of the crossbeam. Due to the limitation of the physical connection between the crossbeam and the sliders of the two parallel guides, the above phenomenon could cause the sliders to deform. This could cause them to wear out or even be damaged [3]. Therefore, precise synchronization is crucial to ensure the motion accuracy of the dual-drive gantry stage. A novel optimal synchronization control algorithm based on model decoupling is proposed to reduce the synchronization error of the dual-drive motors of the gantry stage when it is moving with a heavy-load on the crossbeam.

When the load of the dual-drive motors is unbalanced or subjected to various disturbances, their movement could be out of synchronization. The advanced decoupling control strategy could improve the motion synchronization of the gantry dual-drive motors under different working conditions. Currently, many scholars have proposed various decoupling control strategies. Meng et al. [4] presented a model predictive control strategy. It is based on a switched LTI control-oriented model, which is able to ensure a coordinated contouring tolerance in the presence of disturbances from imperfect drive synchronizations. Yunbo et al. [5] introduced a commercial controller to realize the synchronous control of the dual-drive gantry stage. Ishizaki et al. [6] proposed a cross-coupling PD controller to penalize differential positioning errors between the linear drives of a gantry axis by modifying the reference position and velocity commands. Wang et al. [7] proposed a method that integrates the model’s reference adaptive control and variable structure control. Dongmei et al. [8] presented the decoupling control algorithm with a position controller for improved variable structure control. Sliding-mode variable structure control comprise disturbance rejection, an insensitivity to parameter variations, and simple implementation [9,10]. Kim et al. [11] proposed the LQR optimal control method for improving the synchronous accuracy of gantry dual-diver motors. However, the quantitative modeling of the dual-drive axis’s cross-coupling force is absent in the above controller design approaches.

Xiaoqing et al. [12] proposed a novel fractional-order biquad notch filter and feedforward controller with the inverse model. It generated an antiresonance peak to guarantee the system’s stability and quickly attenuated external disturbances. Qin et al. [13] proposed a synchronous control strategy with the inverse system method. It decoupled thrust forces dynamically through state feedback. Li et al. [14] proposed a synchronous control method for dual-drive systems based on disturbance observers (DOBs). Garcia et al. [15] presented a feedback feedforward decoupling control by model inversion. Tan et al. [16] proposed a DOB-augmented composite control strategy. It coordinates the motion control of the dual-drive gantry stage for precision applications. However the inverse system model method has high requirements for system modeling. As long as there is a deviation in modeling, the zero point of the right half-plane will become the right half-plane’s pole, resulting in a divergence of the system [17].

Chao et al. [2] proposed an adaptive robust synchronization control method by considering the dynamic load presented on the crossbeam. Li et al. [18] proposed the Adaptive Robust Control (ARC) algorithm to obtain a guaranteed robust performance under the presence of uncertain nonlinearities and parametric uncertainties. Cong et al. [1] proposed an ARC algorithm with thrust allocation. It achieved improved motion synchronization in terms of the dual-drive motors and the simultaneous regulation of internal forces. Li et al. [19] presented Desired Compensation Direct/Indirect Adaptive Robust Control (DCDIARC) to synthesize the synchronization controller. It guaranteed both robust performances relative to various uncertainties and accurate parameter estimation. Chen et al. [20] proposed an adaptive model compensation algorithm. It had an accurate online parameter estimation for effectively dealing with uncertain nonlinearities and transformed the contouring tolerance control problem into a robust stabilization problem. However, the above adaptive algorithm needs to proceed through many iterations to converge to the extreme value [21].

Quan et al. [22] proposed a method that transforms the synchronous control problem of such coupled systems into a linear-quadratic optimal control problem. Gomand et al. [23] derived a control structure based on a physical model. However, these algorithms do not consider the change in heavy-load positions on the crossbeam. Aiming at the coupling effect on dual-drive motors subjected to the heavy-load position change, this paper proposed the virtual-centroid-based Gantry Synchronization Linear Quadratic Regulator (GSLQR) optimal control and force-FF decoupling control algorithm. It dramatically reduces the synchronization error of the gantry dual-drive motors due to heavy-load position changes.

The main contributions of this paper are as follows.

(1) An accurate physical model is established to obtain the essential characteristics of the heavy-load dual-drive gantry stage. It includes the crossbeam’s linear motion and rotational motion. The established model clearly shows that the coupling effect of the dual-drive $Y_{1},Y_{2}$ motors is mainly caused by the change in heavy-load $M_{l}$ positions on the crossbeam. Thus, the moment of inertia *J* of the crossbeam could not be described by a constant value. The validity of the established model is confirmed by the actual system.

(2) By establishing a linear quadratic performance index, including synchronization error, the GSLQR optimal control algorithm is designed by using the variational functional extremum method. It derives the control law, including cross-coupling compensation, which preliminarily decouples the system.

(3) By defining the virtual centroid of the crossbeam, the cross-coupling force between dual-drive motors is quantified. Then, the virtual-centroid-based GSLQR optimal control and force-FF decoupling control algorithm are proposed to further improve the synchronization accuracy of the system. Compared with the GSLQR optimal control algorithm and the thrust allocation algorithm in [1], the simulation experiment shows the effectiveness and superiority of the proposed algorithm.

This paper is organized as follows: In Section 2, the lumped parameter dynamics model was established by Newtonian mechanics. Then, a novel decoupling optimal synchronous control algorithm and the relevant control laws are provided. In Section 3, simulation experiments are designed and compared. Finally, a conclusion is drawn in Section 4.

## 2. Materials and Methods

### 2.1. Physical Modeling of Heavy-Load Dual-Drive Gantry System

#### 2.1.1. Equivalent Dynamic Model of the System

The dual-drive axis (*Y*-axis) of the studied industrial dual-drive gantry stage is driven by two PMLMs. Two magnetic rails (PMLM’s stator) are fixed on the marble base in parallel. A precision ball linear-rolling guide rail is installed parallel to each magnetic rail. Two PMLM rotors are rigidly connected with the guide’s sliding block. The crossbeam orthogonal to the parallel guide rail is rigidly connected with the guide’s sliding block. The heavy-load on the crossbeam is driven by a PMLM (*X*-axis). Due to the low rigidity of the sliding block ball’s bearing relative to the joint part, it can be regarded as an elastic element [24]. The finite element of the rigid–flexible coupling characteristics of the dual-drive gantry stage was analyzed in [23]. Based on this, the dynamic model is established for the *Y*-axis’s linear motion and rotational motion of the crossbeam. It generates the equivalent lumped parameter model, as shown in Figure 1. Note, when the rotation angle α of the crossbeam is 0°, the dual-drive motors move synchronously; when α is not 0°, the dual-drive motors move asynchronously. As both joints are subjected to the same angle, they can be modeled by a single equivalent spring with stiffness $k_{\alpha }$.

In Figure 1, OXY represents a fixed inertial coordinate system with the origin *O* located in the middle of the parallel guide rails. The *Y*-axis of OXY is parallel to the linear guide rails $Y_{1}$ and $Y_{2}$. Denote *C* as the equivalent centroid of the crossbeam. $CX^{\prime }Y^{\prime }$ represents the moving inertial coordinate system on the crossbeam. The $X^{\prime }$-axis of $CX^{\prime }Y^{\prime }$ is parallel to the longitudinal direction of the crossbeam. Denote $M_{l}$ as heavy-load on the crossbeam. Denote *M* as the mass of the crossbeam. Denote $y_{1}$ and $y_{2}$ as $Y_{1}$ and $Y_{2}$ motor position values (obtained from the linear grating encoder). Denote $b_{1}$ and $b_{2}$ as the damping coefficients of the linear guide rails. Denote *L* as the length of the crossbeam. Denote $l_{1}$ and $l_{2}$ as the distances between *C* and $Y_{1}$ and $Y_{2}$ motors.

Above all, in the OXY coordinate system, the motion of the crossbeam can be completely described by the following three generalized coordinates: the two position: $x_{C}$, $y_{C}$ of the centroid *C*, and the rotation angle α. From Figure 1, we have the following.
(1)$$y_{C}=y_{1}+l_{1}\sin\alpha =y_{2}-l_{2}\sin\alpha$$

Due to the physical constraints of the gantry stage, α is very small; thus, set α≈0. Thus, sinα=α, and the real-time feedback of $y_{C},\alpha$ can be obtained as follows.
(2)$$\begin{array}{c} y_{C}\;=\;\frac{l_{2}}{L}y_{1}+\frac{l_{1}}{L}y_{2},\\ \alpha \;=\;\frac{1}{L}\left(y_{2}-y_{1}\right) \end{array}$$

Considering that the mass of heavy-load $M_{l}$ relative to the crossbeam cannot be ignored. The movement of $M_{l}$ will cause a change in centroid *C* on the crossbeam. Then, the moment of inertia *J* of the crossbeam cannot be described as a constant value. It can be described as follows.

Suppose a body of mass rotates about an axis passing through its centroid. It has a moment of inertia subject to this axis. Then, the moment of inertia concerning the new axis, parallel to the first axis, can be obtained by the parallel axis theorem [25]. Thus, the following is the case.
(3)$$J=\frac{ML^{2}}{12}\;+\;M{\left(\frac{L}{2}-l_{1}\right)}^{2}$$

Expand (Equation 3) to obtain the following.
(4)$$J=\frac{M}{3}\left(l_{1}^{2}-l_{1}l_{2}+l_{2}^{2}\right)$$

**Remark** **1.**
*The rotational inertia J in *(Equation 4)* is a function of $l_{1}$ and $l_{2}$, unlike the constant set in most paper [1,13,14]. This renders the equivalent model of this paper closer to the actual system.*


From Figure 1, when the dual-drive motors is not synchronized, the *X*-axis and the $X^{\prime }$- axis will form angle α. Equivalent centroid *C* will be changed with the movement of heavy-load $M_{l}$, resulting in the unbalanced load of the $Y_{1}$ and $Y_{2}$ motors.

Above all, using Newtonian dynamics and the PMLM mathematical model [1,13], the equivalent model of the dual-drive gantry stage can be described by: (Equation 5)–(Equation 8)
(5)$$M{\ddot{y}}_{C}=f_{1}+f_{2}-B_{t}{\dot{y}}_{C}+C_{\dot{y}\dot{\alpha }}\dot{\alpha }$$
(6)$$J\ddot{\alpha }=-f_{1}l_{1}+f_{2}l_{2}+C_{\dot{y}\dot{\alpha }}{\dot{y}}_{C}-B_{\alpha }\dot{\alpha }-K_{\alpha }\alpha$$
(7)$$f_{i}=k_{ti}i_{i}$$
(8)$$u_{i}=k_{ei}{\dot{y}}_{i}+L_{ai}{\dot{i}}_{i}+R_{ai}i_{i}$$
where i=1,2; $f_{i}$, $i_{i}$, $u_{i}$, $k_{ti}$, $k_{ei}$, $L_{ai}$, and $R_{ai}$, respectively, represent the motor thrust, coil current, input voltage, motor thrust constant, Back EMF constant, coil inductance, and coil resistance. $B_{t}=b_{1}+b_{2}$, $C_{\dot{y}\dot{\alpha }}=b_{1}l_{1}-b_{2}l_{2}$, $B_{\alpha }=b_{1}l_{1}^{2}+b_{2}l_{2}^{2}$. Expanding (Equation 5) and (Equation 6) yields the following.
(9)$$\frac{Ml_{2}}{L}{\ddot{y}}_{1}+\frac{Ml_{1}}{L}{\ddot{y}}_{2}=f_{1}+f_{2}-b_{1}{\dot{y}}_{1}-b_{2}{\dot{y}}_{2}$$
(10)$$\begin{array}{c} \frac{J}{L}\left({\ddot{y}}_{2}-{\ddot{y}}_{1}\right)=-f_{1}l_{1}+f_{2}l_{2}{\;+\;b}_{1}l_{1}{\dot{y}}_{1}-b_{2}l_{2}{\dot{y}}_{2}-\frac{K_{\alpha }}{L}\left(y_{2}-y_{1}\right) \end{array}$$

#### 2.1.2. Full State-Space Equation of the System

Combined with (Equation 7)–(Equation 10), the state equation of the system could be described as follows:(11)$$\dot{\vec{x}}=A\vec{x}+B\vec{u}$$
(12)$$\vec{y}=C\vec{x}$$
where system state ${\vec{x}}^{T}=\left(y_{1}\;\;y_{2}\;\;{\dot{y}}_{1}\;\;{\dot{y}}_{2}\;\;i_{1}\;\;i_{2}\right)$, system input ${\vec{u}}^{T}=(u_{1}\;\;u_{2})$, and system output ${\vec{y}}^{T}=\left(y_{1}\;\;y_{2}\right)$ is the position value of the $Y_{1},Y_{2}$ motors.
$$A=(\begin{array}{cccccc} 0 & 0 & 1 & 0 & 0 & 0\\ 0 & 0 & 0 & 1 & 0 & 0\\ a_{31} & a_{32} & a_{33} & a_{34} & a_{35} & a_{36}\\ a_{41} & a_{42} & a_{43} & a_{44} & a_{45} & a_{46}\\ 0 & 0 & \frac{-k_{e1}}{L_{a1}} & 0 & \frac{-R_{a1}}{L_{a1}} & 0\\ 0 & 0 & 0 & \frac{-k_{e2}}{L_{a2}} & 0 & \frac{-R_{a2}}{L_{a2}} \end{array}),$$
$$B=(\begin{array}{cc} 0 & 0\\ 0 & 0\\ 0 & 0\\ 0 & 0\\ \frac{1}{L_{a1}} & 0\\ 0 & \frac{1}{L_{a2}} \end{array}),C=(\begin{array}{cccccc} 1 & 0 & 0 & 0 & 0 & 0\\ 0 & 1 & 0 & 0 & 0 & 0 \end{array}).$$

The elements $a_{31}$~$a_{46}$ in the state-space matrix *A* are described by the following.


$$\begin{array}{l} a_{31}=\frac{-K_{\alpha }l_{1}}{Jl_{1}+Jl_{2}},a_{32}=\frac{K_{\alpha }l_{1}}{Jl_{1}+Jl_{2}},a_{33}=\frac{-b_{1}L}{Ml_{1}+Ml_{2}}-\frac{b_{1}Ll_{1}^{2}}{Jl_{1}+Jl_{2}},a_{34}=\frac{-b_{2}L}{Ml_{1}+Ml_{2}}+\frac{b_{2}Ll_{1}l_{2}}{Jl_{1}+Jl_{2}},\\ a_{35}=\frac{k_{t1}L}{Ml_{1}+Ml_{2}}+\frac{k_{t1}Ll_{1}^{2}}{Jl_{1}+Jl_{2}},a_{36}=\frac{k_{t2}L}{Ml_{1}+Ml_{2}}-\frac{k_{t2}Ll_{1}l_{2}}{Jl_{1}+Jl_{2}},a_{41}=\frac{K_{\alpha }l_{2}}{Jl_{1}+Jl_{2}},a_{42}=\frac{-K_{\alpha }l_{2}}{Jl_{1}+Jl_{2}},\\ a_{43}=\frac{-b_{1}L}{Ml_{1}+Ml_{2}}+\frac{b_{1}Ll_{1}l_{2}}{Jl_{1}+Jl_{2}},a_{44}=\frac{-b_{2}L}{Ml_{1}+Ml_{2}}-\frac{b_{2}Ll_{2}^{2}}{Jl_{1}+Jl_{2}},a_{45}=\frac{k_{t1}L}{Ml_{1}+Ml_{2}}-\frac{k_{t1}Ll_{1}l_{2}}{Jl_{1}+Jl_{2}},\\ a_{46}=\frac{k_{t2}L}{Ml_{1}+Ml_{2}}+\frac{k_{t2}Ll_{2}^{2}}{Jl_{1}+Jl_{2}}. \end{array}$$


The relevant parameters of the system could be found in the reference manual of the machine manufacturer and are provided in Table 1.

Substituting the parameters of Table 1 into state-space matrix *A*, with related coefficients being $l_{1}=0.2$ m and $l_{2}=0.6$ m, yields the following.



$\begin{array}{c} A=(\begin{array}{cccccc} 0 & 0 & 1 & 0 & 0 & 0\\ 0 & 0 & 0 & 1 & 0 & 0\\ -5627 & 5627 & -0.286 & 0.057 & 3.486 & -0.697\\ 16881 & -16881 & 0.057 & -0.971 & -0.697 & 11.85\\ 0 & 0 & -9783 & 0 & -1657 & 0\\ 0 & 0 & 0 & -9783 & 0 & -1657 \end{array}) \end{array}$



By conducting MATLAB calculations, the controllable matrix of the system $S_{c}=(A\;\;AB\;\;\ldots \;\;A^{5}B)$ is at full rank. Thus, the system is controllable.

#### 2.1.3. Validity of the Established Model

Note that the state-space matrix *A* is strongly coupled. From [23], the gantry stage is a square MIMO (multiple input multiple output system). It can be described as follows: (13)$$\left(\begin{array}{c} {\dot{y}}_{1}\\ {\dot{y}}_{2} \end{array})=(\begin{array}{cc} G_{11}\left(s\right) & G_{12}\left(s\right)\\ G_{21}\left(s\right) & G_{22}\left(s\right) \end{array})(\begin{array}{c} u_{1}\\ u_{2} \end{array}\right)$$
where $G_{11}\left(s\right)$ and $G_{22}\left(s\right)$ are the direct transfers of $Y_{1},Y_{2}$ motors, and $G_{12}\left(s\right)$ and $G_{21}\left(s\right)$ are the cross-coupling transfer functions between $Y_{1}$ and $Y_{2}$ motors.

From [26], state space Equations (Equation 11) and (Equation 12) are written in the form of a transfer function, which yields the following.
(14)$$\begin{array}{l} \frac{Y}{U}=C{\left(sI-A\right)}^{-1}B=\\ \;\;\left(\begin{array}{cccccc} 1 & 0 & 0 & 0 & 0 & 0\\ 0 & 1 & 0 & 0 & 0 & 0 \end{array}\right){\left(sI-A\right)}^{-1}(\begin{array}{cc} 0 & 0\\ 0 & 0\\ 0 & 0\\ 0 & 0\\ \frac{1}{L_{a1}} & 0\\ 0 & \frac{1}{L_{a2}} \end{array}) \end{array}$$

Thus, the following is the case.
(15)$$\begin{array}{cc} G_{11}\left(s\right)=\frac{{\dot{y}}_{1}}{u_{1}}{|}_{u_{2}=0} & =(\begin{array}{cccccc} 0 & 0 & 1 & 0 & 0 & 0 \end{array}){\left(sI-A\right)}^{-1}\left(\begin{array}{c} 0\\ 0\\ 0\\ 0\\ \frac{1}{L_{a1}}\\ 0 \end{array}\right)\\ & =\frac{687.5s^{4}+1.14\times 10^{6}s^{3}+9.07\times 10^{7}s^{2}+1.795\times 10^{10}s}{s^{6}+3315s^{5}+2.922\times 10^{6}s^{4}+3.267\times 10^{8}s^{3}+6.691\times 10^{10}s^{2}+1.805\times 10^{12}s+0.2184} \end{array}$$
(16)$$\begin{array}{cc} G_{22}\left(s\right)=\frac{{\dot{y}}_{2}}{u_{2}}{|}_{u1=0} & =\left(\begin{array}{cccccc} 0 & 0 & 0 & 1 & 0 & 0 \end{array}\right){\left(sI-A\right)}^{-1}\left(\begin{array}{c} 0\\ 0\\ 0\\ 0\\ 0\\ \frac{1}{L_{a2}} \end{array}\right)\\ & =\frac{2338s^{4}+3.874\times 10^{6}s^{3}+9.07\times 10^{7}s^{2}+1.795\times 10^{10}s}{s^{6}+3315s^{5}+2.922\times 10^{6}s^{4}+3.267\times 10^{8}s^{3}+6.691\times 10^{10}s^{2}+1.805\times 10^{12}s+0.2184} \end{array}$$
(17)$$\begin{array}{cc} G_{12}\left(s\right)=G_{21}\left(s\right) & =\frac{{\dot{y}}_{1}}{u_{2}}{{|}_{u_{1}=0}=\frac{{\dot{y}}_{2}}{u_{1}}|}_{u_{2}=0}=\left(\begin{array}{cccccc} 0 & 0 & 0 & 1 & 0 & 0 \end{array}\right){\left(sI-A\right)}^{-1}\left(\begin{array}{c} 0\\ 0\\ 0\\ 0\\ \frac{1}{L_{a1}}\\ 0 \end{array}\right)\\ & =\frac{-137.5s^{4}-2.278\times 10^{5}s^{3}+1.083\times 10^{7}s^{2}+1.795\times 10^{10}s}{s^{6}+3315s^{5}+2.922\times 10^{6}s^{4}+3.267\times 10^{8}s^{3}+6.691\times 10^{10}s^{2}+1.805\times 10^{12}s+0.2184} \end{array}$$

The experimental stage of the system is shown in Figure 2. When the heavy-load is close to the $Y_{1}$ motor side, centered, and close to the $Y_{2}$ motor side, the relevant frequency characteristic curves of the system are calculated or tested, as shown in Figure 3, Figure 4, Figure 5, Figure 6, Figure 7 and Figure 8. The open-loop frequency characteristic curve of the equivalent model is shown in Figure 3, Figure 5 and Figure 7. The closed-loop frequency characteristic curve of the actual stage is obtained by sweeping the frequency of the motors, as shown in Figure 4, Figure 6 and Figure 8. Compared with Figure 3 and Figure 4, Figure 5 and Figure 6, and Figure 7 and Figure 8, bode diagrams of both the model and the actual system show that the frequency response performance of the light-load motor is better than that of the heavy load motorl they have similar characteristics in the low-frequency band (the main working frequency band of this gantry stage, 100∼1000 rad/s). By conducting this simple comparison experiment, it can be confirmed that the established model can be used for the algorithm comparison simulation experiment in the following paper. If the experimental verification of the algorithm is carried out in the future, the least square method satisfying the PE condition is needed to accurately identify the parameters of the actual system.

### 2.2. Virtual-Centroid-Based GSLQR Optimal Control and Force-FF Decoupling Control Algorithm Design

To reduce the synchronization error of the dual-drive motors in the gantry stage with the dynamic heavy-load $M_{l}$, the following two objectives will be completed in this section: 1. The GSLQR optimal control algorithm is designed to preliminarily compensate for the cross-coupling force of the system; 2. to further optimize $Y_{1}$ and $Y_{2}$ motors’ synchronization accuracy, the virtual-centroid-based GSLQR optimal control and force-FF decoupling control algorithm is proposed.

#### 2.2.1. GSLQR Optimal Control Algorithm Design

The linear-quadratic-regulator (LQR) control law $\vec{u}\;=\;-k\vec{x}$ is designed to minimize $I=\lim_{t\to \infty }I\left(t\right)$ in (Equation 18). Generally, with the setting of weight matrix *Q* of the system state and input weight *R*, the optimal feedback gain *k* could be calculated by the Riccati equation.
(18)$$\begin{array}{cc} \; & I=\int_{0}^{t_{f}}\frac{1}{2}\left({\vec{x}}^{T}Q\vec{x}+{\vec{u}}^{T}R\vec{u}\right)d\tau {+\vec{x}(t}_{f}{)}^{T}Q_{f}{\vec{x}(t}_{f})\\ & \;s.t.\;\dot{\vec{x}}=A\vec{x}+B\vec{u} \end{array}$$

To realize the design of the Gantry Synchronization Linear Quadratic Regulator (GSLQR) optimal control algorithm, the quadratic performance index of synchronization errors between dual-drive motors should be introduced. The detailed procedure is described below.

To guarantee the synchronous movement of the $Y_{1},Y_{2}$ motors, the position values $y_{1},y_{2}$ should always keep the minimum error when the dual-drive axis moves; thus, $y_{1},y_{2}$ and the desired trajectory $y_{d}$ should be as consistent as possible. The desired state is set as ${\vec{x}}_{d}^{T}=\left(y_{d}\;\;y_{d}\;\;0\;\;0\;\;0\;\;0\right)$. By defining $\varepsilon_{1}\;=\;y_{1}-y_{d}$, $\varepsilon_{2}\;=\;y_{2}-y_{d}$, the system state is rewritten as $\vec{\varepsilon }=\vec{x}-{\vec{x}}_{d}$. Thus, ${\vec{\varepsilon }}^{T}=\left(\varepsilon_{1}\;\;\varepsilon_{2}\;\;{\dot{y}}_{1}\;\;{\dot{y}}_{2}\;\;i_{1}\;\;i_{2}\right)$.

Since, the the system state at the terminal time $t_{f}$ is 0. Substituting $\vec{x}$ by $\vec{\varepsilon }$ in (Equation 18) yields the following.
(19)$$I=\int_{0}^{t_{f}}\frac{1}{2}\left({\vec{\varepsilon }}^{T}Q\vec{\varepsilon }+{\vec{u}}^{T}R\vec{u}\right)d\tau$$

To guarantee (Equation 19), obtain the global minimum under the constraint condition of (Equation 11). The following equivalent convex function by introducing the Lagrange multiplier $\vec{\lambda }$ [27] should be constructed.
(20)$$\begin{array}{cc} \; & I_{\mathrm{convex}}=\\ & \int_{0}^{t_{f}}\left[\frac{1}{2}\left({\vec{\varepsilon }}^{T}Q\vec{\varepsilon }+{\vec{u}}^{T}R\vec{u}\right)+{\vec{\lambda }}^{T}(A\vec{x}+B\vec{u}-\dot{\vec{x}})\right]d\tau \end{array}$$

Define $\ell \;=\;\frac{1}{2}\left({\vec{\varepsilon }}^{T}Q\vec{\varepsilon }+{\vec{u}}^{T}R\vec{u}\right)$; taking the total variation of $I_{\mathrm{convex}}$ in (Equation 20) yields the following:(21)$$\begin{array}{cc} \; & \delta I_{\mathrm{convex}}=\\ & \int_{0}^{t_{f}}\left(\frac{\partial \ell }{\partial \vec{\varepsilon }}\delta \vec{\varepsilon }+\frac{\partial \ell }{\partial \vec{u}}\delta \vec{u}+{\vec{\lambda }}^{T}A\delta \vec{x}+{\vec{\lambda }}^{T}B\delta \vec{u}-{\vec{\lambda }}^{T}\delta \dot{\vec{x}}\right)d\tau \end{array}$$
where $\frac{\partial \ell }{\partial \vec{\varepsilon }}\;=\;{\vec{\varepsilon }}^{T}Q$, $\frac{\partial \ell }{\partial \vec{u}}={\vec{u}}^{T}R$, $\delta \vec{\varepsilon }\;=\;\delta \left(\vec{x}-{\vec{x}}_{d}\right)$. Note that ${\vec{x}}_{d}$ is a fixed trajectory; then, $\delta {\vec{x}}_{d}\;=\;0$; thus, $\delta \vec{\varepsilon }\;=\;\delta \vec{x}$ (if define as $\vec{\varepsilon }={\vec{x}}_{d}-\vec{x}$, then $\delta \vec{\varepsilon }\;=\;-\delta \vec{x}$, which will not obtain (Equation 29)). The last term in (Equation 21) can be modified using integration by the following parts:(22)$$-\int_{0}^{t_{f}}\lambda^{T}\delta \dot{x}d\tau =-\lambda^{T}\left(t_{f}\right)\delta x\left(t_{f}\right)+\lambda^{T}\left(0\right)\delta x\left(0\right)+\int_{0}^{t_{f}}{\dot{\lambda }}^{T}\delta xd\tau$$
where $\lambda^{T}\left(0\right)\delta x\left(0\right)=0$. From (Equation 21) and (Equation 22), we obtain the following:(23)$$\begin{array}{cc} \delta I_{\mathrm{convex}}= & \int_{0}^{t_{f}}\left({\vec{\varepsilon }}^{T}Q+{\vec{\lambda }}^{T}A+{\dot{\vec{\lambda }}}^{T}\right)\delta \vec{x}d\tau +\\ & \int_{0}^{t_{f}}\left({\vec{u}}^{T}R+{\vec{\lambda }}^{T}B\right)\delta \vec{u}d\tau -{\vec{\lambda }}^{T}\left(t_{f}\right)\delta \vec{x}\left(t_{f}\right) \end{array}$$
where $\delta \vec{x}$, $\delta \vec{u}$, and $\delta \vec{x}\left(t_{f}\right)\in R$. To obtain an optimal control solution that minimizes $I_{\mathrm{convex}}$, the following three terms must be equal to 0.
(24)$${\vec{\varepsilon }}^{T}Q+{\vec{\lambda }}^{T}A+{\dot{\vec{\lambda }}}^{T}\;=\;0$$
(25)$${\vec{u}}^{T}R+{\vec{\lambda }}^{T}B\;=\;0$$
(26)$$-{\vec{\lambda }}^{T}\left(t_{f}\right)\;=\;0$$

Note that constraint (Equation 26) represents an initial condition for the reverse-time equation for $\vec{\lambda }$ starting at $t_{f}$. Thus, the dynamics in (Equation 11) with initial condition ${\vec{x}\left(0\right)=\vec{x}}_{0}$ and (Equation 24)–(Equation 26) with the final-time condition form a two-point boundary value problem. Since the system could be approximated as a linear system, it is possible to assume that $\vec{\lambda }\;=\;P\vec{\varepsilon }$. Combing (Equation 24) yields the following.
(27)$${\vec{\varepsilon }}^{T}Q+{\left(P\vec{\varepsilon }\right)}^{T}A+{\left(\dot{P}\vec{\varepsilon }+P\dot{\vec{\varepsilon }}\right)}^{T}\;=\;0$$

Transposing and expanding (Equation 27) yields the following.
(28)$$Q\vec{\varepsilon }+A^{T}P\vec{\varepsilon }+\dot{P}\vec{\varepsilon }+PA\vec{\varepsilon }-PBR^{-1}B^{T}P\vec{\varepsilon }\;=\;0$$

When t→∞, $\dot{P}=0$, we obtain the following Algebraic Riccati Equation (ARE).
(29)$$Q+A^{T}P+PA-PBR^{-1}B^{T}P\;=\;0$$

An optimal solution to the *P* matrix can be obtained by using backward approximate dynamic programming. Substituting the optimal solution *P* into (Equation 25) can obtain the optimal feedback gain $k_{GSLQR}=R^{-1}B^{T}P$. Thus, the control law is obtained as follows.
(30)$$\vec{u}\;=\;-R^{-1}B^{T}P\vec{\varepsilon }$$

From(Equation 29), both GSLQR and LQR have robustness, as described as follows: The LQR achieves infinite gain margin $k_{g}=\infty$ and also guarantees phase margin $p=60^{\circ }$. It was proved by Lyapunov’s second method in [28,29].

#### 2.2.2. Virtual-Centroid-Based Force-FF Decoupling Control Algorithm Design

As observed from Figure 3, the DC components of the coupling term $G_{12}\left(s\right)$ and the $G_{11}\left(s\right)$ and $G_{22}\left(s\right)$ are almost the same. To improve the synchronization accuracy of the dual-drive gantry stage, the effect of the coupling term on the system must be reduced. Since the GSLQR optimal control algorithm is robust, adding force-FF decoupling to the GSLQR could further improve the synchronization accuracy of the system [30,31,32].

In [33], Richard et al. proposed the concept of inverse models of causal-order graphs. By its methodology, this paper proposed the virtual-centroid-based GSLQR optimal control and force-FF decoupling control algorithm to further improve the synchronous accuracy of the system.

Defining the virtual centroid $\frac{M}{2}{\ddot{y}}_{1},\frac{M}{2}{\ddot{y}}_{2}$ and rewriting (Equation 9)–(Equation 10) as control-oriented equations yields the following.
(31)$$Eq\_vm1:f_{1}-f_{bc1}-f_{xc}=\frac{M}{2}{\ddot{y}}_{1}+b_{1}{\dot{y}}_{1}$$
(32)$$Eq\_vm2:f_{2}-f_{bc2}+f_{xc}=\frac{M}{2}{\ddot{y}}_{2}+b_{2}{\dot{y}}_{2}$$



$\begin{array}{c} Eq\_coupled:\\ \left\{\begin{array}{c} f_{bc1}=\frac{K_{\alpha }}{L^{2}}\left(y_{1}-y_{2}\right)+\frac{J-Ml_{1}l_{2}}{L^{2}}\left({\ddot{y}}_{1}-{\ddot{y}}_{2}\right)-\frac{M(l_{1}-l_{2})}{2L}{\ddot{y}}_{1}\\ f_{bc2}=-\frac{K_{\alpha }}{L^{2}}\left(y_{1}-y_{2}\right)-\frac{J-Ml_{1}l_{2}}{L^{2}}\left({\ddot{y}}_{1}-{\ddot{y}}_{2}\right)+\frac{M(l_{1}-l_{2})}{2L}{\ddot{y}}_{2} \end{array}\right. \end{array}$



In (Equation 31)–(Equation 32), $f_{bc1}$ and $f_{bc2}$ are cross-coupling forces of unbalanced load relative to $Y_{1},Y_{2}$ motors (the movement of heavy-load $M_{l}$ will cause centroid *C* to change in the longitudinal direction of the crossbeam). $f_{xc}$ is the cross-coupling force of the eccentric load to $Y_{1},Y_{2}$ motors.

Thus, the block diagram of the proposed algorithm is shown in Figure 9.

${\tilde{y}}_{1},\tilde{\ddot{y_{1}}},{\tilde{l}}_{1},{\tilde{y}}_{2},\tilde{\ddot{y_{2}}},{\tilde{l}}_{2},{\tilde{f}}_{1},{\tilde{f}}_{2},{\tilde{f}}_{bc1}$, and ${\tilde{f}}_{bc2}$ can be obtained directly or indirectly through the linear grating position encoder of the dual-drive gantry stage.

From Figure 9, the decoupling force ${\tilde{f}}_{bc1},{\tilde{f}}_{bc2}$ is added into the system’s control loop as force-feedforward.

## 3. Simulation Experiments

The proposed algorithm experiment is carried out in the Matlab/Simulink environment, and it will be compared with the GSLQR optimal control algorithm and the thrust allocation algorithm ($u_{1}/u_{2}=k_{m}l_{2}/l_{1}=k_{t2}l_{2}/k_{t1}l_{1}$) presented in [1].

The simulation experiment of the GSLQR optimal control algorithm will be carried out first. To penalize the synchronization error, $\varepsilon_{1}$ and $\varepsilon_{2}$ in system state ${\vec{\varepsilon }}^{T}=(\varepsilon_{1}\;\;\varepsilon_{2}\;\;{\dot{y}}_{1}\;\;{\dot{y}}_{2}\;\;i_{1}\;\;i_{2})$ require high weight values. Thus, the weights are set as Q=diag(5005001111), R=1. Then, the optimal feedback gain $k_{GSLQR}$ can be obtained by Matlab calculations. With the position command of (Equation 33), the $Y_{1},Y_{2}$ motor’s response speed curve is shown in Figure 10.
(33)$$y_{d}=\left\{\begin{array}{cc} 2.5t^{2}, & 0<t<0.2\\ t-0.1, & 0.2⩽t<2\\ 2.4, & t⩾2 \end{array}\right.$$

The simulation results show that when load $M_{l}$ is located at different positions of the crossbeam, the GSLQR optimal feedback control can ensure that the response speeds of the $Y_{1},Y_{2}$ motors are basically the same. However, as obesrved from Figure 10a,c, when the load of the dual-drive motors becomes more and more unbalanced, the synchronization errors of the $Y_{1},Y_{2}$ motors tend to deteriorate. It should be pointed out that the effectiveness of the linear quadratic optimal control algorithm is mainly due to the precondition of (Equation 1) (α≈0; thus, sinα=α). Thanks to the rigid connection between the crossbeam and the $Y_{1},Y_{2}$ guide rails, it can always ensure α≈0 when the stage is in motion; that is, the motion of the stage is generally dominated by linear equations. In practical operation, the range of α is also related to the clearance of the guide: The larger the clearance, the larger α. If α≈0 is not satisfied, a non-linear solution needs to be developed, which will be carried out in the next study.

To further improve the synchronization accuracy of $Y_{1},Y_{2}$ motors, the force-FF decoupling will be added in the following experiments.

The position of heavy-load $M_{l}$ is adjusted to the $Y_{1}$ motor side by setting $l_{1}=0.2$ m, $l_{2}=0.6$ m. The weights are set as Q=diag(500500110.10.1) and R=0.02. The optimal feedback gain can be obtained as follows: $k_{GSLQR}=(\begin{array}{cccccc} 79.6373 & 78.4503 & 4.4432 & 1.3175 & 0.3010 & 0.0073\\ 85.4521 & 72.6880 & 3.9487 & 1.7949 & 0.0073 & 0.3033 \end{array})$.

The force-FF gain of decoupling force ${\tilde{f}}_{bc1},{\tilde{f}}_{bc2}$ is set to 0.13 by experience. With the position step command of (Equation 34), the synchronization error curve of each algorithm is shown in Figure 11.
(34)$$y_{d}=\left\{\begin{array}{cc} 0, & 0<t<1\\ 0.1, & 1⩽t<2\\ 0, & t⩾2 \end{array}\right.$$

The maximum value of synchronization error of each algorithm in Figure 11 is shown in Table 2.

The position of heavy-load $M_{l}$ is adjusted to the $Y_{2}$ motor side by setting $l_{1}=0.5$ m and $l_{2}=0.3$ m. The weights are set as Q=diag(500500110.10.1), R=0.02. The optimal feedback gain could be obtained as follow: $k_{GSLQR}=\left(\begin{array}{cccccc} 72.5974 & 85.5296 & 2.4536 & 3.2945 & 0.3034 & 0.0066\\ 81.2974 & 76.8034 & 1.9788 & 3.7780 & 0.0066 & 0.3017 \end{array}\right.$.

The force-FF gain of decoupling force ${\tilde{f}}_{bc1},{\tilde{f}}_{bc2}$ is set to 0.13. With the position step command of (Equation 34), the synchronization error curve of each algorithm is shown in Figure 12.

The maximum value of synchronization error of each algorithm in Figure 12 is shown in Table 3.

From the above simulation results, when the heavy-load $M_{l}$ is located at different positions of the crossbeam, the proposed algorithm is compared with the GSLQR optimal control algorithm and the thrust allocation algorithm in [1] and reduces the maximum synchronization error by about 70 % and 60 %, respectively.

## 4. Conclusions

The issue of the coupling effect on dual-drive motors subjected to heavy-load position changes has been investigated. Aiming at this problem, the virtual-centroid-based GSLQR optimal feedback control and force-feedforward decoupling control algorithm is proposed. The simulation results show that the proposed algorithm greatly improves the synchronization accuracy of the dual-drive motors.

In this paper, an accurate mathematical model was established for the dual-drive gantry stage with dynamic heavy-load, which includes the linear motion and rotational motion of the crossbeam. Unlike most gantry stage rotation dynamics, which consider the crossbeam rotational inertia *J* as a constant, the rotational inertia *J* in this paper is a function of the centroid’s position variable *C*. The validity of the model is confirmed by the frequency response identification experiment of the actual system. The model shows that the coupling effect of the dual-drive $Y_{1},Y_{2}$ motors is mainly caused by the change of heavy-load $M_{l}$’s position on the crossbeam.

From the model, the strongly coupled state-space matrix of the system is obtained. Unlike configuring PID to independently control $Y_{1}$ and $Y_{2}$ motors, this paper proposes the Gantry Synchronous Quadratic Linear Regulation (GSLQR) optimal algorithm to control the dual-drive axis in one system. The cross-coupling force of the system is preliminarily compensated by the optimal feedback algorithm. The systematic design procedure of the controller and its robustness have been clearly presented.

To further improve the synchronization accuracy of the $Y_{1},Y_{2}$ motors, a virtual centroid is defined to quantify the cross-coupling force between dual-drive motors. The force-feedforward decoupling control is added to further compensate for the cross-coupling force of the system. Unlike the thrust allocation algorithm [1], which only focuses on the centroid position variable *C*, the quantized coupling force derived in this paper reveals that the coupling of the dual-drive motor is mainly generated by the different accelerations of the $Y_{1}$ and $Y_{2}$ motors.

The simulation results show the effectiveness and superiority of the virtual-centroid-based GSLQR optimal feedback control and force-feedforward decoupling control algorithm: compared with the thrust allocation algorithm in [1], the maximum synchronization error is reduced by about 60%.

## Figures and Tables

**Figure 1 entropy-24-01153-f001:**
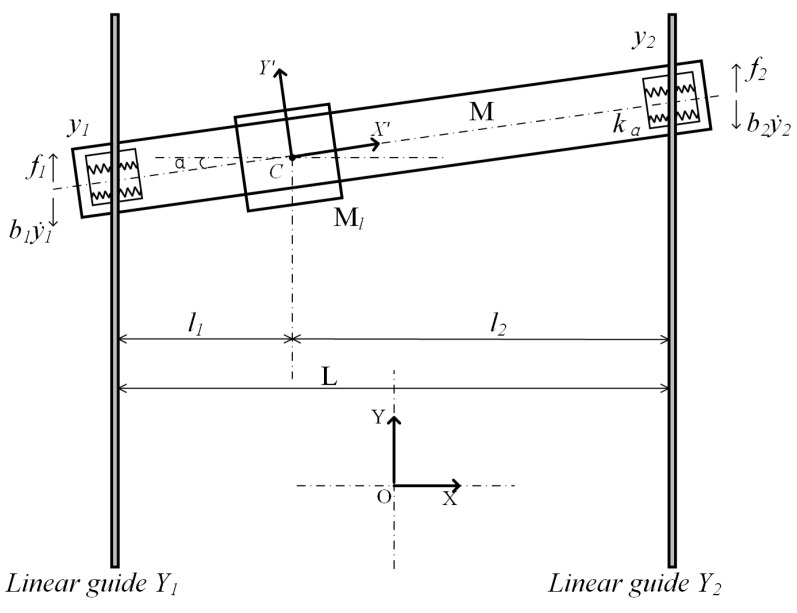
Equivalent lumped parameter model of a dual-drive gantry stage.

**Figure 2 entropy-24-01153-f002:**
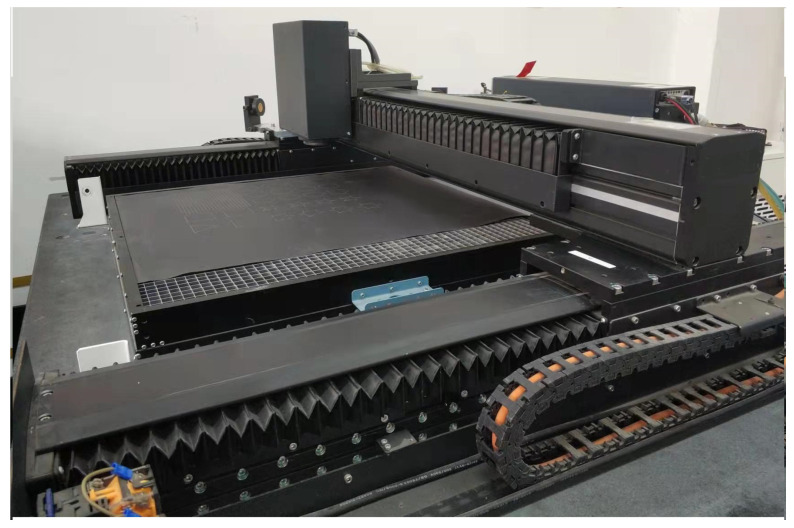
Dual-drive gantry stage.

**Figure 3 entropy-24-01153-f003:**
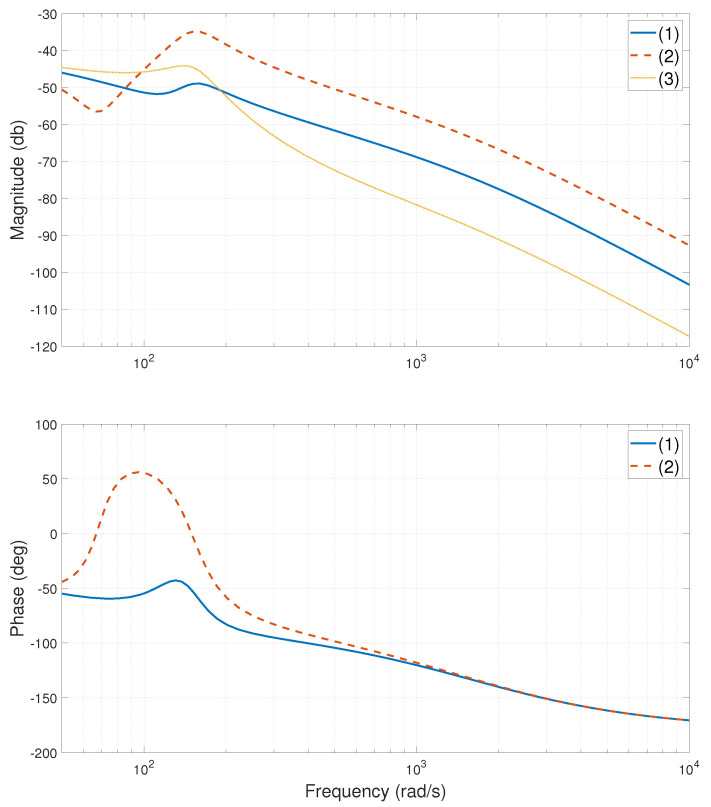
Bode diagram of model (open loop), heavy-load close $Y_{1}$ motor side: (1)—$G_{11}\left(s\right)$; (2)—$G_{22}\left(s\right)$; (3)—$G_{12}\left(s\right)$.

**Figure 4 entropy-24-01153-f004:**
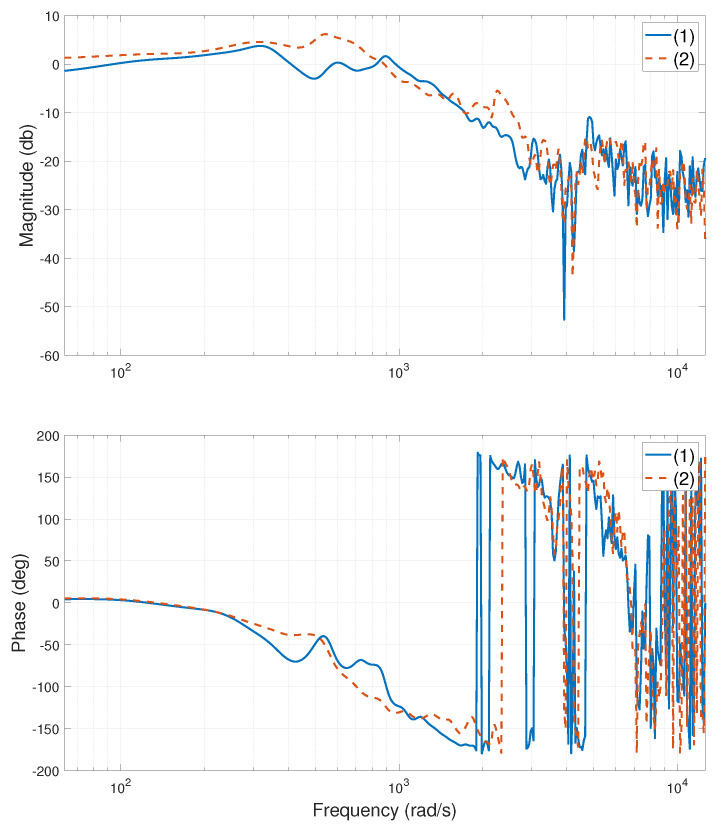
Bode diagram of actual system (closed loop), heavy-load close $Y_{1}$ motor side: (1)—$Y_{1}$ motor; (2)—$Y_{2}$ motor.

**Figure 5 entropy-24-01153-f005:**
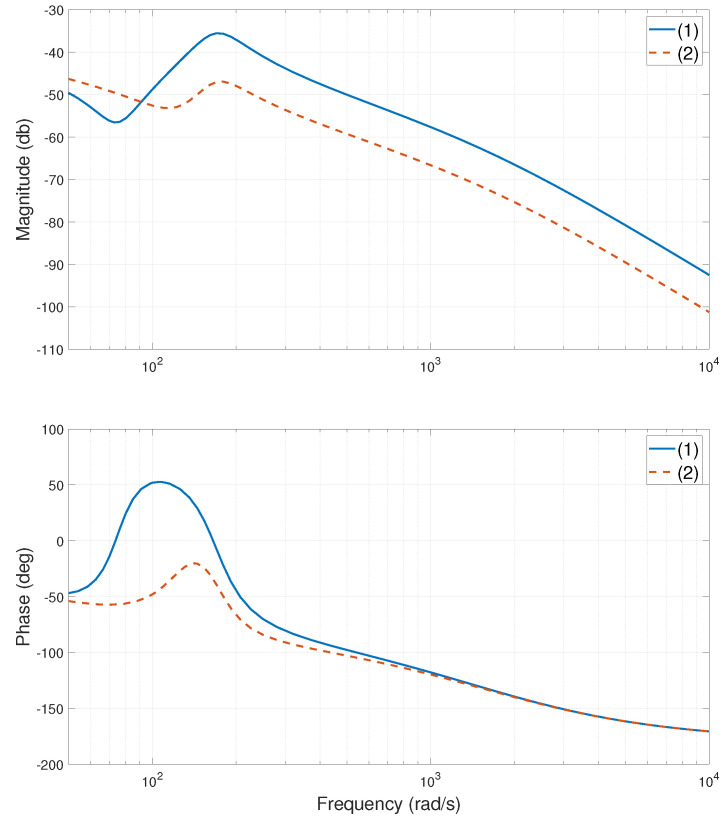
Bode diagram of model (open loop), heavy-load close $Y_{2}$ motor side: (1)—$G_{11}\left(s\right)$; (2)—$G_{22}\left(s\right)$.

**Figure 6 entropy-24-01153-f006:**
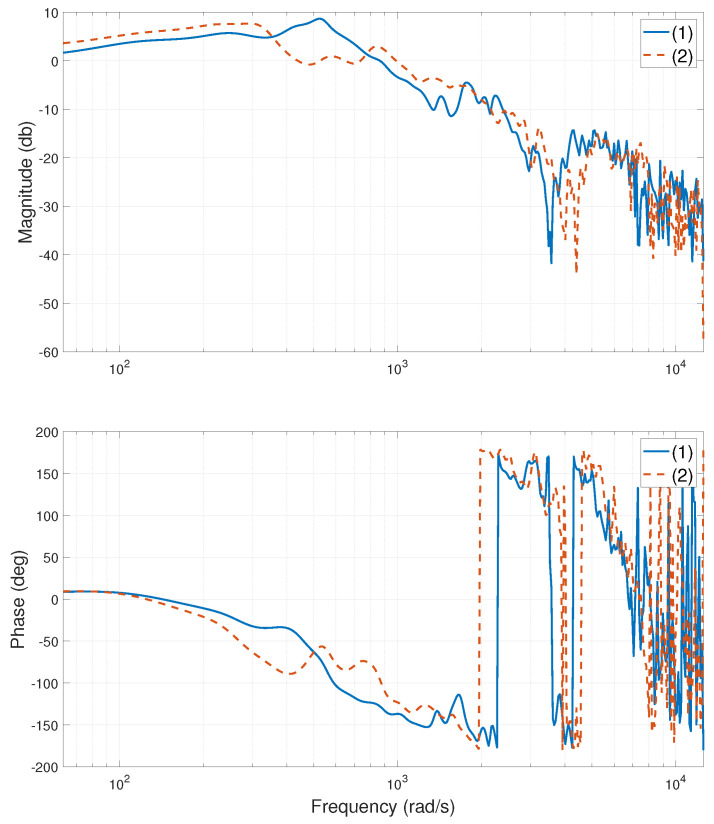
Bode diagram of actual system (closed loop), heavy-load close $Y_{2}$ motor side: (1)—$Y_{1}$ motor; (2)—$Y_{2}$ motor.

**Figure 7 entropy-24-01153-f007:**
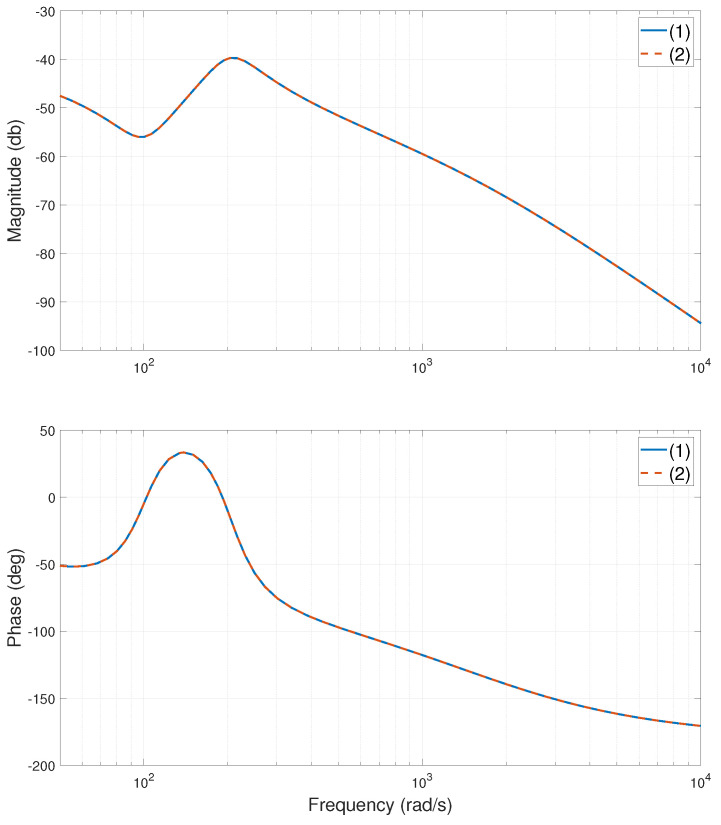
Bode diagram of model (open loop), heavy-load centered. (1)—$G_{11}\left(s\right)$; (2)—$G_{22}\left(s\right)$.

**Figure 8 entropy-24-01153-f008:**
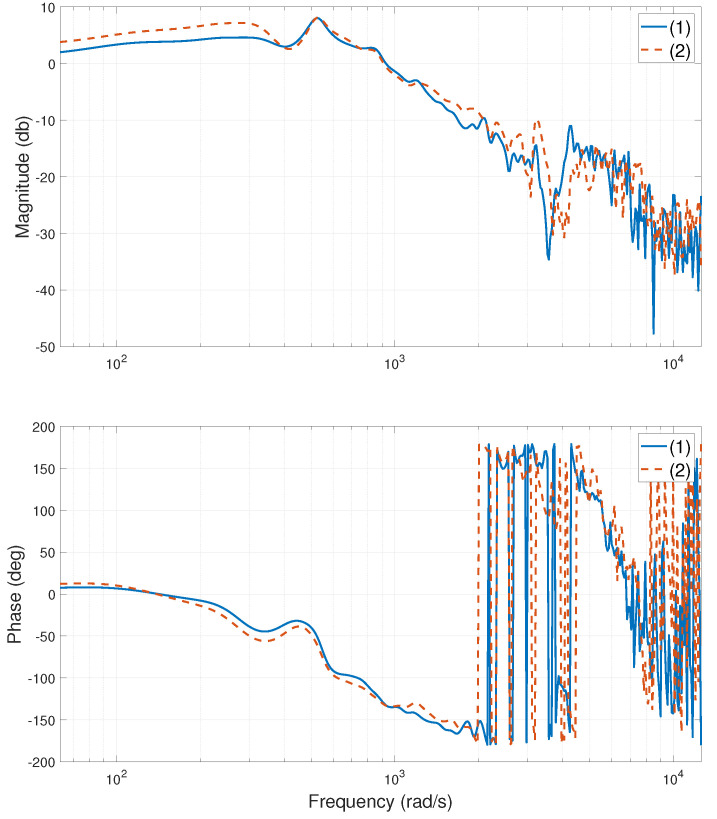
Bode diagram of actual system (closed loop), heavy-load centered. (1)—$Y_{1}$ motor; (2)—$Y_{2}$ motor.

**Figure 9 entropy-24-01153-f009:**
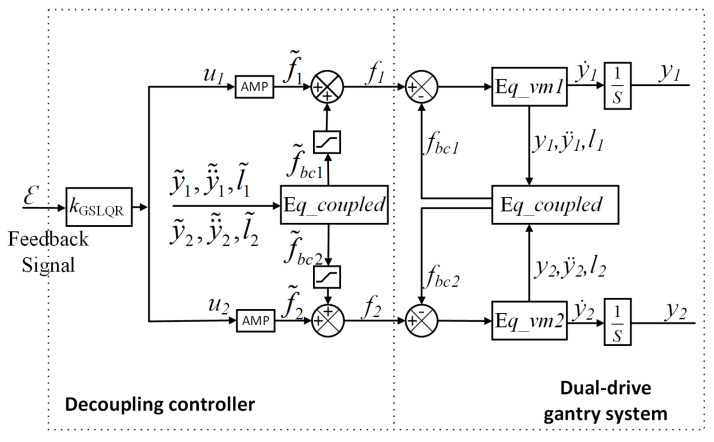
Block diagram of the virtual-centroid-based GSLQR optimal control and force-FF decoupling control algorithm. AMP referred to an amplifier.

**Figure 10 entropy-24-01153-f010:**
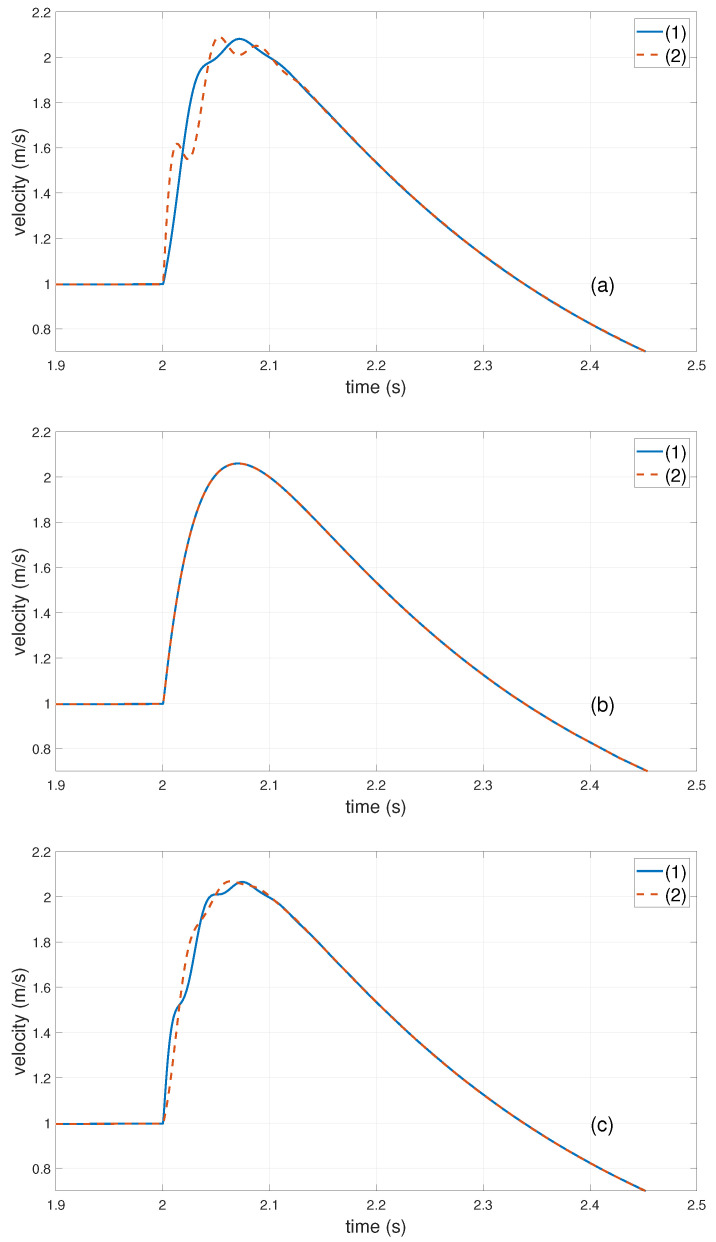
Simulation result of GSLQR optimal control algorithm: (**a**) the load $M_{l}$ is close to $Y_{1}$ motor side, $l_{1}=0.2$ m, $l_{2}=0.6$ m; (**b**) the load $M_{l}$ is centered, $l_{1}=l_{2}=0.4$ m; (**c**) the load $M_{l}$ close to $Y_{2}$ motor side, $l_{1}=0.5$ m, $l_{2}=0.3$ m. (1)—speed of $Y_{1}$ motor; (2)—speed of $Y_{2}$ motor.

**Figure 11 entropy-24-01153-f011:**
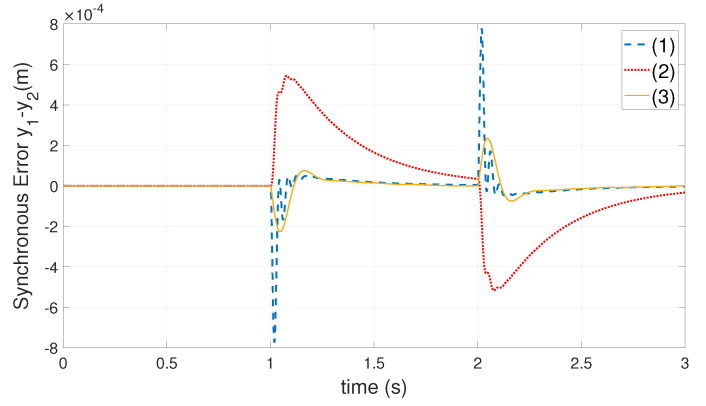
The synchronization error curve of each algorithm: (1)—GSLQR optimal control algorithm; (2)—thrust allocation algorithm [1]; (3)—proposed algorithm. Where heavy-load $M_{l}$ close to the $Y_{1}$ motor side, $l_{1}=0.2\;m,l_{2}=0.6\;m$.

**Figure 12 entropy-24-01153-f012:**
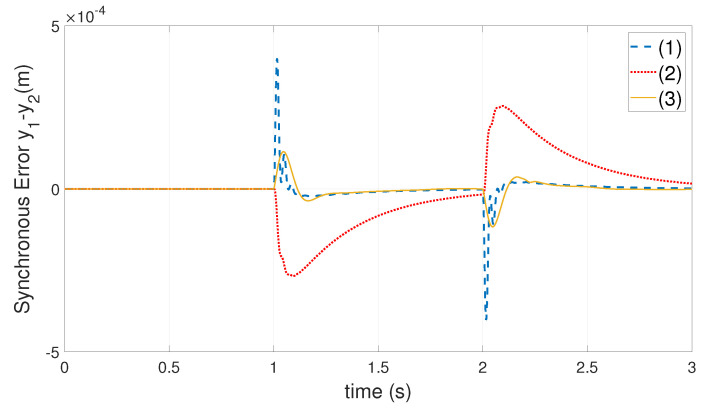
The synchronization error curve of each algorithm: (1)—GSLQR optimal control algorithm; (2)—thrust allocation algorithm [1]; (3)—proposed algorithm. Where the heavy-load $M_{l}$ close to the $Y_{2}$ motor side, $l_{1}=0.5\;m,l_{2}=0.3\;m$.

**Table 1 entropy-24-01153-t001:** The relevant parameters of the system.

Name	Symbol	Value
midrule mass of crossbeam (including load $M_{l}$)	*M*	25 kg
mass of load	$M_{l}$	10 kg
length of crossbeam	*L*	0.8 m
damping of $Y_{1}$ rotor	$b_{1}$	5 N·m·s
damping of $Y_{2}$ rotor	$b_{2}$	5 N·m·s
stiffness of joint between crossbeam and rails	$K_{\alpha }$	52,520 N/m
thrust constant of $Y_{1}$ Motor	$k_{t1}$	61.0 N/A
thrust constant of $Y_{2}$ Motor	$k_{t2}$	61.0 N/A
back EMF constant of $Y_{1}$ motor	$k_{e1}$	49.6 V/M/S
back EMF constant of $Y_{2}$ motor	$k_{e2}$	49.6 V/M/S
inductance of $Y_{1}$ Motor	$L_{a1}$	$5.07\times 10^{-3}$ H
inductance of $Y_{2}$ Motor	$L_{a2}$	$5.07\times 10^{-3}$ H
resistance of $Y_{1}$ Motor	$R_{a1}$	8.4 Ω
resistance of $Y_{2}$ Motor	$R_{a2}$	8.4 Ω

**Table 2 entropy-24-01153-t002:** The heavy-load $M_{l}$ close to the $Y_{1}$ motor side, the maximum value of the synchronization error of each algorithm: (1)—GSLQR optimal control algorithm; (2)—thrust allocation algorithm [1]; (3)—proposed algorithm.

Index	Algo (1)	Algo (2)	Algo (3)
$\max\left(\left|y_{1}-y_{2}\right|\right)$, mm	0.76	0.54	0.22

**Table 3 entropy-24-01153-t003:** The heavy-load $M_{l}$ close to the $Y_{2}$ motor side, the maximum value of the synchronization error of each algorithm: (1)—GSLQR optimal control algorithm; (2)—thrust allocation algorithm [1]; (3)—proposed algorithm.

Index	Algo (1)	Algo (2)	Algo (3)
$\max\left(\left|y_{1}-y_{2}\right|\right)$, mm	0.39	0.26	0.11

## Data Availability

The data that support the findings of this study are available from the corresponding author upon reasonable request.

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
