# Peer review of "Modeling Dual-Drive Gantry Stages with Heavy-Load and Optimal Synchronous Controls with Force-Feed-Forward Decoupling"

_entropy, 2022, doi:10.3390/e24081153_

Round 1

Reviewer 1 Report

This paper presents the synchronisation control method for dual drive machines with simulation results. The authors are expected to answer the following questions and revise the paper accordingly:

1. The system description is not clear.  Some parameters are not defined properly, like 'M_l' in line 132. Some parameters are not defined but used in the equations, like f_i, i_i in eq (5)-(8).

2. The proposed GSLQR is to minimize the synchronization error. But the designed cost function doesn't penalize the difference between y1 and y2, how does it work?

3. How do you choose the value of system parameters as Table 1 on page 6 should be discussed.

4. Since there is no system identification and model validation, how do you conclude the statement ‘it can be confirmed that the established model can be used for the algorithm comparison’?

5. According to eq(9) and eq(10), this work presents a similar model as shown in [1] where the movement along the X’ axis is ignored. Is this assumption applicable in real applications? 

6. The double derivatives of y1 and y2 are very noisy and may cause problems when feedforwarding these terms. This may not be obvious in simulation but can be severe in real applications. How do you solve this?

Also, the manuscript should be double checked to avoid the following flaws:

- In the abstract, we do not use symbols. The symbol J on line 9 can be deleted.

- The abbreviation that appeared in the abstract should be defined in the manuscript as well such as GSLQR.

- No need to capitalize ‘Using’ at the line under 146. Similar issues should be avoided in the whole manuscript.

- what is ‘?’ in the citation on line 214

- what is ‘liner guide’ in Figure 1?

Reviewer 2 Report

Thearticle dealing with Modeling of Daul Drive in the described application is very interesting because of the usage of FF decoupling algorithm.

The introduction part is quite broad, however, some references dealing with mechanism simulations can be also used by comparing the authors methods, presented e.g. in Hrcek et al. - Global Sensitivity Analysis of Chosen Harmonic Drive Parameters Affecting Its Lost Motion.

The part 2 - Materials and methods describes the authors approach clear enough to understand the proposed equations used in the model of the dual-drive gantry stage.

I really appreciate the experimental part on the real gantry stage presented in Fig. 2. The GSLQR algorithm is described adequately, followed by the simulation experiments.

Formal reminder - please check the spaces below the Figure captions, which are not present by the Figures - it makes the text a little bit not clear.

Round 2

Reviewer 1 Report

N.A

Author Response

N.A